# Development of Lutein-Containing Eye Drops for the Treatment of Dry Eye Syndrome

**DOI:** 10.3390/pharmaceutics13111801

**Published:** 2021-10-27

**Authors:** Yi-Zhou Chen, Zhi-Yu Chen, Yu-Jun Tang, Cheng-Han Tsai, Yu-Lun Chuang, Erh-Hsuan Hsieh, Lachlan Tucker, I-Chan Lin, Ching-Li Tseng

**Affiliations:** 1Graduate Institute of Biomedical Materials and Tissue Engineering, College of Biomedical Engineering, Taipei Medical University, No. 250, Wu-Hsing Street, Taipei City 11031, Taiwan; max4617@hotmail.com (Y.-Z.C.); d79010340217@gmail.com (Z.-Y.C.); darkduck77@hotmail.com (Y.-J.T.); joyo4180@gmail.com (C.-H.T.); a3750585@gmail.com (Y.-L.C.); haish711228@hotmail.com (E.-H.H.); lachlan.tucker@utas.edu.au (L.T.); 2Menzies Institute for Medical Research, University of Tasmania, Hobart, TAS 7000, Australia; 3Department of Ophthalmology, Taipei Medical University Shuang Ho Hospital, New Taipei City 23561, Taiwan; 4Department of Ophthalmology, School of Medicine, College of Medicine, Taipei Medical University, Taipei City 11031, Taiwan; 5College of Biomedical Engineering, Taipei Medical University, Taipei City 11031, Taiwan; 6College of Medicine, Taipei Medical University, Taipei City 11031, Taiwan; 7Research Center of Biomedical Device, College of Biomedical Engineering, Taipei Medical University, Taipei City 11031, Taiwan

**Keywords:** lutein, eye drops, dry eye syndrome, anti-inflammation, polyvinyl alcohol

## Abstract

Dry eye syndrome (DES) is a common ophthalmological disease that decreases tear secretion and causes dryness, photophobia, pain, severe corneal rupture, and even blindness. Ocular and lacrimal gland inflammation is one of the pathological mechanisms underlying DES. Therefore, effective suppression of inflammation is a crucial strategy for the treatment of DES. Lutein, commonly found in healthy foods, has anti-inflammatory effects in corneal or retina-related cells and may be a potential therapy for DES. The addition of lutein to artificial tears (AT) as an eye-drop formulation for DES treatment in a mouse model was studied in the present work. Polyvinyl alcohol (PVA) was used as a thickener to increase the viscosity of eye drops to prolong drug retention on the ocular surface. A WST-8 assay in human corneal epithelial cells (HCE-2) showed that a concentration of <5 μM lutein (L5) and <1% PVA (P1) maintained the cell viability at 80%. A real-time PCR showed that the inflamed human corneal epithelial cells (HCECs) cocultured with L5P1 had downregulated expression of inflammatory genes such as IL-1β, IL-6, and TNF-α. In a benzalkonium chloride- (BAC) induced DES mouse model, AT/L5P1 could repair damaged corneas, elevate tear secretion, increase the number of goblet cells, and inhibit the production of inflammatory cytokines, such as IL-1β, IL-6, and TNF-α, in the cornea. In conclusion, we demonstrate that lutein/PVA as eye drops could prolong the drug ocular retention time and effectively to decrease inflammation in DES mice. Therefore, lutein, obtained from eye drops, has a potential therapeutic role for DES.

## 1. Introduction

Dry eye syndrome (DES) is one of the most common eye diseases. DES is a multifactorial disease associated with tear film instability, visual disturbance, and potential ocular surface damage, as defined by the International Dry Eye Workshop (DEWS) [1]. Factors that cause DES include age, sex, disease history, medications, lifestyle changes, and environmental conditions [2,3,4,5,6]. Approximately 100 million people worldwide suffer from DES, including 10–15% (50–60 million) of adults in the United States [7,8]. According to recent research in Taiwan, the prevalence of DES in the population over 65 years of age is as high as 33.7% [9]. The pathogenesis of DES has recently been extensively documented in relation to the hyperosmolarity of tears and inflammation of the ocular surface [10,11,12,13]. Overexpression of inflammatory cytokines, including interleukin (IL)-6, IL-1β, IL-17, chemokine (C-C motif) ligand 2 (CCL2), interferon-γ (IFN-γ), and tumor necrosis factor-alpha (TNF-α), causes goblet cell loss, leading to tear film instability and tear hyperosmolality [13,14,15,16]. The instability of the tear film causes apoptosis of the corneal epithelial cells, leading to an increase in the expression of inflammatory cytokines in the ocular surface, creating a vicious circle [1,17,18].

Artificial tears (AT) are the most commonly used agents for DES treatment at the early DES stage. Although treatment with AT could improve the symptoms quickly, it was ineffective in relieving the inflammatory cycle of DES described above [13,14,15,16]. Flavonoids from plants with anti-inflammatory effects were tested in DES rabbits; inflammatory cytokines in the cornea, such as IL-6, IL-8, IL-1β, and TNFα, were reduced, and tear volume increased. The polyphenol component selected for these experiments was epigallocatechin gallate (EGCG)—the primary component of green tea, and kaempferol—found in *Ginkgo biloba* [19,20]. 

Lutein is one of the most common flavonoid components in dark green, leafy vegetables, such as kale and spinach [21]. It has good antioxidation and anti-inflammatory properties, which can suppress mammalian tumor growth and enhance immune function [22]. Lutein has also been identified as one of the retinal macular components and is present at a high level—the reason for many healthy foods containing lutein claiming a retinal maintenance function [23]. Lutein can reduce inflammatory levels of NF-κB, IL-1β, and Cox-2 in cultured Müller cells after hypoxic injury [24]. Chao et al. also showed that lutein could effectively reduce the expression of hyperosmotic-induced secretion of IL-6 in human corneal epithelial cells (HCECs) through the deactivation of p38, JNK, and NF-κB pathways [25]. Therefore, lutein inhibits inflammatory effects in eye-related cells (Müller and HCECs).

Polyvinyl alcohol (PVA) is a synthetic polymer widely used in the medical field with good biocompatibility, chemical resistance, and high water solubility [26]. Medical products, including artificial cartilage, maintenance fluids for contact lenses, and artificial vitreous compositions, contain PVA [27]. A previous study showed that PVA and hyaluronic acid (HA) were added to AT as thickening agents, both of which had similar effects [28]. The price of PVA is much lower than HA, making it a competitive raw material, compared to HA, for financial reasons. Adding PVA to AT both thickens the solution and improves the dispersion on the ocular surface to enhance the retention of active components on the ocular surface [29].

In this study, eye drops containing lutein were chosen as the active component for treating the inflammatory condition in DES animals, and PVA was added as the lubricant for enhancing the drug (lutein) retention on the eye. The anti-inflammatory capacity was evaluated in vitro using lipopolysaccharide- (LPS) induced inflammation in human corneal epithelial cells (HCE-2). The therapeutic effect of lutein eye drops was investigated in a mouse model of DES.

## 2. Materials and Methods

### 2.1. Materials and Reagents

Lutein was acquired from USBiological Life Science (Salem, MA, USA). PVA, hydrocortisone, Triton™ X-100, 4′,6-diamidino-2-phenylindole (DAPI), cell counting Kit-8 (CCK-8), live/dead cell double staining kit, lipopolysaccharide, benzalkonium chloride (BAC), and dimethyl sulfoxide (DMSO) were purchased from Sigma-Aldrich (St. Louis, MO, USA). Epidermal growth factor (EGF) was purchased from PeproTech (Rocky Hill, NJ, USA). Keratinocyte serum-free medium (KSFM), bovine pituitary extract (BPE), insulin, trypsin-EDTA, and penicillin/streptomycin were purchased from Gibco BRL (Gaithersburg, MD, USA). Tetramethylrhodamine succinyl ester (TAMRA-SE) and TRIzol reagent were obtained from Invitrogen (Carlsbad, CA, USA). FNC coating mix (containing fibronectin, collagen, and albumin) was obtained from Athena Environmental Sciences, Inc. (Baltimore, MD, USA). High-capacity complementary DNA (cDNA) Reverse Transcription Kit and TaqMan Fast Universal Master Mix (2×) were purchased from Applied Biosystems (Foster City, CA, USA). Rompun solution (2%) was obtained from Bayer Korea, Ltd. (Ansan-city, Korea), and Zoletil 50 was purchased from Virbac Animal Health (Vauvert, Nice, France). Fluorescein paper strips were obtained from HAAGSTREIT AG (Koniz, Switzerland). Topical anesthesia solution (Alcaine^®®^ 0.5% ophthalmic solution) was obtained from Alcon-Couvreur N.V. (Puurs, Belgium). Zone-Quick phenol red cotton thread was obtained from Menicon (Tokyo, Japan). All other chemicals were purchased from Sigma-Aldrich.

### 2.2. Cytotoxicity of HCE-2 Treated with PVA or Lutein 

#### 2.2.1. Cell Culture

The human corneal epithelial cell line HCE-2 was purchased from the American Type Culture Collection (No. CRL-11135; Manassas, VA, USA), and KSFM with 5 ng/mL EGF, 5 ng/mL insulin, 50 ng/mL BPE, and 500 ng/mL hydrocortisone was used as culture media. HCE-2 cells were subcultured and maintained at 37 °C in a 5% CO_2_ incubator. Before HCE-2 seeding, tissue culture plastics were precoated with an FNC coating mix solution for 30–60 s. 

#### 2.2.2. Cell Viability Examination

The viability of HCE-2 cells treated with various concentrations of lutein or PVA was determined using the CCK-8 assay kit. Lutein was dissolved in DMSO at a high concentration as a stock solution and then diluted in culture medium for testing. HCE-2 cells were seeded in 96-well plates at a density of 5 × 10^3^ cells per well and cultured overnight. After that, the HCE-2 cells were cocultured with lutein at different concentrations (1.25–40 μM) or PVA at 0.01–2% (*v*/*v*) for 1 and 3 days. The culture medium was discarded, and 0.1 mL of a working solution of WST-8 was added to each well. After 2 h of incubation at 37 °C, the absorbance was measured at 450 nm using a microplate reader (Multiskan GO; Thermo Fisher Scientific, Waltham, MA, USA).

#### 2.2.3. Live/Dead Staining

HCE-2 cells were seeded in 24-well plates (3 × 10^4^ cells/well) in culture medium and incubated overnight. HCE-2 cells were treated with various lutein concentrations and 1% PVA, which was derived from the results of the cell viability examination in the previous step, and were stained with a live/dead staining kit (04511-1KT-F, Sigma-Aldrich) to observe live cells. Cells with green and red fluorescence were viable and dead, respectively. Images were acquired using an inverted fluorescence microscope (IX81; Olympus, Tokyo, Japan).

### 2.3. Gene Expression of Inflammatory Cytokines in HCE-2

HCE-2 cells were seeded in 6-well plates (3 × 10^5^ cells/well) in culture medium and incubated overnight. The medium was replaced with a fresh medium containing 500 ng/mL LPS-containing fresh medium. Non-LPS-treated cells were used as controls. After LPS stimulation for 6 h, the medium was removed and treated with fresh medium containing 1% PVA (abbreviated as P1), 5 μM lutein (L5), 10 μM lutein (L10), 5 μM lutein mixed 1% PVA (L5P1), and 10 μM lutein mixed with 1% PVA (L10P1) for 2 h. Cells were collected, and the total amount of RNA was extracted using TRIzol reagent. The first strand of complementary DNA was synthesized from 0.2 μg/μL RNA using a high-capacity cDNA Reverse Transcription Kit. Real-time PCR was performed using a StepOne Real-Time PCR System (Applied Biosystems) with the TaqMan Universal PCR Master Mix (2×) and the following primers: IL-1β (Hs01555413m1), TNFα (Hs00174128m1), and IL-6 (Hs00174131m1). GAPDH (Hs99999905m1) was used as an internal control. Relative gene expression was quantified using the ΔΔC_t_ method.

### 2.4. Characterization of AT Mixed with PVA and Lutein

The basal components of the AT solution (100 mL) included 450 mg of NaCl, 150 mg of KCl, 15 mg of CaCl_2_, and 450 mg of Na_2_HPO_4_. The AT solution was freshly prepared, and it was preservative-free and aseptic after filtration through a 0.22 μm filter. In this study, the pH, osmotic pressure, viscosity, and refractive index (RI) were used to assess the AT mixed with PVA and lutein. The pH value was measured using a pH meter (pH 510; Eutech Instruments, Singapore). Osmolarity was determined using a micro-osmometer (Model 3320; Advanced Instruments, Norwood, MA, USA). The RI of the AT mixture was measured using a refractometer (DR-A1 ATAGO, Kyoto, Japan).

### 2.5. Analysis of Ocular Retention Time

Male C57BL/6J mice aged 6–8 weeks were used to examine the ocular retention time of the AT mixture. All experimental procedures were approved by the Institutional Animal Care and Use Committee of Taipei Medical University (approval no. LAC-2017-0395, 19 March 2018). The animals were housed in standard cages in a light-controlled room at 23 ± 2 °C, relative humidity of 60% ± 10%, and alternating 12 h light-dark cycles (6 AM to 6 PM). Each animal was provided food and water ad libitum. TAMRA fluorescent dye (2 μg/mL) was added to three AT mixtures (AT, AT/L5, AT/L5P1), and 2 μL was dropped onto the mouse eye. Xenogen in vivo imaging system (IVIS) (Alameda, CA, USA) was used to observe the fluorescence-retention status of the AT mixture from 10 s to 90 min, and quantitative analysis with software was used to calculate the fluorescence intensity of the AT mixture on the ocular surface.

### 2.6. In Vivo Evaluation Therapeutic Effect of AT Mixed with PVA and Lutein by DES Mice Model

Sixty C57BL/6J male mice aged 6–8 weeks were used in this study for the DES mouse model. First, 0.1% BAC was administered twice daily for 13 days to induce mice with DES, as previously described, with slight modification [30]. Tear volume secretion and corneal fluorescein staining were examined before and after BAC treatment to confirm DES induction. Mice were randomly divided into six groups and treated with different eye drops: (1) normal (without any induction and treatment), (2) DES (0.1% BAC, negative control), (3) cyclosporin A (CsA, RESTASIS Ophthalmic Emulsion with 0.05% CsA), (4) AT, (5) AT/L5, and (6) AT/L5P1. Different eye drops (20 µL) were dropped on mouse eyes 2 times daily for 10 days. The mice were euthanized, and their corneas were carefully dissected after the treatment period. A detailed examination of DES conditions is described as follows:

#### 2.6.1. Tear Secretion Evaluation and Fluorescein Staining

Tear volume was measured using a Zone-Quick phenol red cotton thread [30]. Briefly, after the mice were anesthetized and following topical administration of 0.5% Alcaine^®^ eye drops, the Zone-Quick cotton thread was placed on the outside of the mouse’s eyes as Schirmer’s test to evaluate tear volume. After 20 s, the thread became red because of the adsorption of tears, and the length of red thread was scored using a Vernier caliper. For corneal fluorescein staining, 2 μL of 1% fluorescein solution was dropped onto the conjunctival sac of the mice. After 90 s, a cotton swab was used to absorb excess dye around the eye, which was then recorded under a slit-lamp microscope with a cobalt blue filter. When the corneal epithelium layer was damaged, green fluorescent dye deposition on the cornea was observed under a slit lamp. 

#### 2.6.2. Hematoxylin and Eosin (H&E) Staining and Periodic Acid-Schiff (PAS) Staining

After 10 days of treatment, the mice were sacrificed, the whole eyeballs were dissected and fixed in 10% buffered formalin solution for 24 h. The fixed specimens were embedded in paraffin and sectioned. The sections were stained with hematoxylin, eosin and periodic acid-Schiff (PAS) stain for histological examination. Each section was observed using an optical microscope (BM-1A; SAGE vision, New Taipei City, Taiwan).

#### 2.6.3. Quantification of Inflammatory Cytokines in Mouse Corneas

The cornea was weighed and separately chopped into small pieces for protein extraction. The cornea was frozen by immersion in liquid nitrogen and then ground with lysis buffer (Tissue Protein Extraction Cocktail, ThermoFisher Scientific, Inc., Waltham, MA, USA). The tissue extract was collected, and the protein content was quantified using a Coomassie protein assay (ThermoFisher Scientific). The protein concentration of the samples was adjusted to the required concentration (30 μg total protein in 200 μL) for the subsequent ELISA assay. The inflammatory cytokine (IL-1β, IL-6, TNF-α) content was determined by a Mouse Cytokine/Chemokine Magnetic Bead Panel 96-well plate assay (MCYTOMAG-70K, Millipore, Darmstadt, Germany).

### 2.7. Statistical Analysis

All data are presented as the mean ± standard deviation (SD) from two to three independent experiments. Statistical differences between groups were analyzed by one-way ANOVA, followed by Tukey’s post hoc test using SPSS 17.0 (SPSS, Inc., Chicago, IL, USA). Statistical significance was set at *p* < 0.05.

## 3. Results

### 3.1. Cytotoxicity of PVA and Lutein in Human Corneal Epithelial Cells

A WST-8 assay was used to evaluate the viability of HCE-2 treated with varying concentrations of PVA or lutein to determine the suitable concentration. No toxicity was observed after one day of culture in the presence of lutein at concentrations from 1.25 to 10 μM; however, significant cytotoxicity occurred above 10 μM. After three days of cultivation, the cell viability of the 10 μM lutein group decreased to less than 50%, showing slight cytotoxicity (Figure 1A). For the PVA test, no toxicity was observed after one day and three days of cultivation in the presence of 0.1–1% PVA, and the cell viability in the 2% PVA-treated group dropped to 80% on the third day (Figure 1B). Since eye drops stay on the ocular surface for a short time (no more than 3 days), we selected lutein (<10 µM) mixed with 1% PVA (P1) for further tests based on one day of cell culture.

The cell viability of HCECs treated with 1% PVA mixed with different concentrations of lutein is shown in Figure 1C. Different concentrations (1.25–10 µM) of lutein mixed with 1% PVA were cultured with HCE-2 cells for 1 and 3 days and did not show cytotoxicity on day 1. The cells treated with lutein 5 μM/P1 maintained 80% cell viability on day 3. Therefore, the optimal concentration for the combination of lutein and PVA for further experiments was determined as 5 μM lutein plus 1% PVA. 

As shown in Figure 2A, no difference in the number of green-stained live cells was found after one day of culture. However, after 3 days of culture, the number of viable cells in the 5 and 10 μM lutein groups mixed with 1% PVA (L5P1, L10P1) significantly decreased, as observed by the live/dead staining results. The number of surviving cells decreased to <20% in the L10P1 group (Figure 2B).

### 3.2. Gene Expression of Inflamed HCECs Treated with AT Mixture

During inflammation, gene expression of IL-6, IL-1β, and TNF-α is usually upregulated. Therefore, we examined the anti-inflammatory effect of various lutein/PVA combinations on LPS-stimulated HCE-2 cells. As shown in Figure 3, 1% PVA alone did not effectively downregulate the expression of IL-6, IL-1β, and TNF-α in HCE-2 cells, showing no inherent anti-inflammatory effect. In the lutein group, both 5 μM (L5) and 10 μM (L10) showed significant downregulation of IL-6 and TNF-α but had no significant effect on IL-1β. However, when L5 and L10 were mixed with 1% PVA (L5P1, L10P1), IL-6, TNF-α, and IL-1β gene expression were significantly decreased. Based on the results of cytotoxicity tests (Figure 1 and Figure 2) and gene expression (Figure 3) results, we found that the safe concentration of lutein/PVA mixture for cells with good anti-inflammatory effects was 5 µM lutein plus 1% PVA.

### 3.3. Characterization of AT Mixed with Lutein and PVA as Eye Drops

The pH values of various AT/lutein/PVA mixtures ranged from 7.78 to 8.37, and the AT/L5P1 pH value was 7.78 ± 0.01 (Table 1). Although pH values were slightly higher than normal human tears (6.5 to 7.6), it is acceptable for eye drops, especially the AT/L5P1. The osmotic pressure and viscosity values of AT/L5P1 were measured as 271 ± 4 mOsm/kg and 1.21 ± 0.02 mPa·s, which matched the normal human tear osmotic pressure (260–340 mOsm/kg) and viscosity range (1–10 mPa·s). The results of RI in all the tested groups were around 1.33, showing the addition of lutein (L5) and PVA (1%) did not influence vision.

### 3.4. Ocular Retention Time of AT Mixed with Lutein and PVA

TAMRA (fluorescent dye) was added to three different AT mixture groups (AT, AT/L5, AT/L5P1) to determine the effect of PVA on the ocular surface. The results of the IVIS imaging system are shown in Figure 4. The fluorescent spots on the eye of AT/L5P1-treated mice can be observed after 90 min (Figure 4A). Approximately 75% (72% ± 7%) of the residual fluorescence of the AT/L5P1 group remained on the ocular surface, compared with the AT and AT/L5 groups, while the residual fluorescence of the AT group and AT/L5 dropped to less than 40% (Figure 4B). This indicates that 1% PVA addition could increase the retention time of eye drops on the eye surface.

### 3.5. Therapeutic Efficacy of Lutein/PVA-Mixed Eye Drops in the DES Mice Model 

#### 3.5.1. Appearance of the Eyeball and Changes in Tear

There were no significant changes in the appearance of the ocular surface (such as discharge, redness, chemosis, or angiogenesis) of all the mice observed upon examination with a slit lamp. Only the DES group showed slight pits in the cornea (Figure 5A). The Schirmer test results are shown in Figure 5B, indicating that the tear secretion volume was decreased in the DES group, compared with that in the normal group, after continuous BAC induction. The CsA and AT group showed just slightly increased tear volume, and the tear volume of the AT/L5P1 group was similar to that of the control group (Figure 5B).

#### 3.5.2. Evaluation of the Repair Effect of the Cornea by Fluorescence and Histological Stain 

After 10 days of treatment, the ocular surface was stained with fluorescein and examined using a slit-lamp microscope. Fluorescent dye penetrated and was deposited on the non-intact corneal epithelium, allowing visual evaluation. The result showed that intense green fluorescein staining was observed in the DES and AT groups. A low level of fluorescence was observed in the normal, CsA, AT/L5, and AT/L5P1 groups after 10 days of treatment (Figure 6A). Quantitation of the fluorescent staining was performed according to the instructions of the National Eye Institute/Industry to evaluate corneal staining [35]. The scores range from 0 to a maximum of 15, with higher scores indicating more damage to the cornea. The fluorescent staining score, as shown in Figure 6B, of the DES group (8.1 ± 1.4) was significantly different from that of the control group (3.3 ± 1.5), demonstrating the successful establishment of the DES model. Treatment with AT alone showed no therapeutic effect; the score was around 7 ± 1.5, which was not different from that of the DES group. The CsA group (Restasis, one of the clinic’s agents to treat DES) revealed less green staining on the cornea (3.4 ± 1.5) (* *p* < 0.05 compared with the DES group). The score of the AT/L5P1 group after 10 days of treatment was 2.9 ± 1.4, which was also statistically different from the DES group (* *p* < 0.05). 

H&E staining (Figure 7A) and quantification of the corneal epithelial thickness (Figure 7B) were performed. The corneal epithelium of normal mice included 3–5 layers of cells with a thickness of 40.4 ± 1.2 μm. In the DES group (0.1% BAC treated one), the number of layers was less, and the thickness was decreased to 16.3 ± 1.2 μm, showing a significant difference with control one (@ *p* < 0.05). After 10 days of treatment, the corneal epithelial thickness of the AT group was 25.1 ± 1.5 μm, while the CsA and AT/L5P1-treated groups were 31.5 ± 1.1 μm and 34.2 ± 0.6 μm, respectively (Figure 7B), which were significantly different from the DES group (* *p* < 0.05). Corneal epithelial thickness was not different between the AT/L5P1 and control groups.

Previous studies have indicated that DES can lead to the loss of goblet cells [36,37]. Assessing the density or number of goblet cells is also an indicator for evaluating DES. The PAS staining results showed that many goblet cells could be observed on the conjunctiva of mice in the normal group, but many were lost in the DES group. After 10 days of treatment, goblet cells were not found in the AT group, but some goblet cells could be observed in the tissue sections of the CsA, AT/L5, and AT/L5P1 groups (Figure 8).

#### 3.5.3. Quantification of Inflammatory Cytokines in Mouse Corneas

The mouse corneas were extracted, and the variation in inflammatory cytokines was examined to evaluate the treatment efficacy of the various lutein/PVA mixtures (Figure 9). In the DES group, the concentration of inflammatory factors such as IL-1β (4.1 ± 0.5 pg/mL), IL-6 (22.7 ± 8.2 pg/mL), and TNF-α (0.2 ± 0.1 pg/mL) was significantly increased, compared with the normal group (@ *p* < 0.05). The concentration of IL-1β (3.4 ± 1.2 pg/mL) and IL-6 (21.5 ± 3.7 pg/mL) was high in the AT group, too, with no significant difference from the DES group. In the CsA group, the TNF-α content (0.08 ± 0.03 pg/mL) was significantly decreased (* *p* < 0.05 compared with DES), while IL-1β (2.9 ± 0.2 pg/mL) and IL-6 (16.6 ± 2.9 pg/mL) showed no difference from the DES group. In the AT/L5P1 group, the concentrations of IL-1β (1.8 ± 0.2 pg/mL), IL-6 (11.7 ± 1.7 pg/mL), and TNF-α (0.07 ± 0.02 pg/mL) were significantly lower compared to those in the DES group (* *p* < 0.05). Based on these results, the AT solution mixed with 5 μM lutein and 1% PVA (AT/L5P1) effectively inhibited inflammation in BAC-induced DES in mice.

## 4. Discussion

Dry eye syndrome is a common eye disease worldwide and affects vision-related quality of life. Investigation of DES is necessary to determine its pathogenesis and to design appropriate therapies. Recent studies of humans and animal models have indicated that topical anti-inflammatory treatment is effective, such as mixing anti-inflammatory substances, including steroids, catechins, ferulic acid, and traditional herbal extracts, into artificial tear solution (AT) or buffer. The goal of this study was to investigate whether lutein, a common ingredient in foods healthy for the eye, is effective for reducing inflammation in DES and relieving DES symptoms by increasing tear volume and healing the damaged cornea upon topical delivery as eye drops.

Lutein is obtained from a healthy diet or nutritional supplements since mammals are not able to synthesize it. Lutein is generally ingested orally as tablets, capsules, or drinks. Parker et al. showed that lutein was absorbed by the mucosa of the small bowel and then secreted into the lymph before reaching the liver [38]. Lutein is then incorporated into lipoproteins that are distributed to peripheral tissues, particularly in the retina [39]. However, lutein absorption through the digestive system may be disrupted by many components, such as fiber, ions, other carotenoids, and even drugs [40]. On the other hand, Obana et al. showed that the total volume of macular pigment optical density significantly increased by week 8 of continuous supplementation with lutein [41]. Oral administration of lutein may take a long time to reach and accumulate in the eye. In this study, we developed a new method of absorbing lutein directly from the cornea through eye drops and investigated its potential for DES treatment.

Since lutein is insoluble in water, it is usually dissolved in an organic solvent, such as ethanol or DMSO, or prepared in nano-assemblies [42]. Gruenert et al. reported that intraocular surgical dyes containing lutein have less cytotoxic effects on the corneal endothelium than traditional trypan blue-based dye [43]. Chao et al. suggested that lutein concentrations of less than 10 μM did not cause cytotoxic effects on corneal epithelial cells and uveal melanocytes [25,44]. Oh et al. also demonstrated the safety of lutein up to 30 μM [45]. The highest lutein concentration tested here (40 μM) contained 0.2% DMSO. Our preliminary test confirmed that < 1% DMSO was nontoxic (cell viability > 80%) for HCE-2 cells. Various concentrations of lutein and the thickener, PVA, were used to test cell viability and cytotoxicity in HCE-2 cells. At 5 μM, lutein alone or mixed with 1% PVA did not reduce cell viability after 1-day treatment. The cell viability was also maintained at 80% after 3 days of cultivation (Figure 1A,C). The PVA concentration of less than 1% did not influence HCE-2 cells’ viability (Figure 1B). Results from live/dead staining also show that the number of living cells was around 70% in the L5P1 group on day 3 (Figure 2). Therefore, the optimal concentration of lutein was 5 μM to use alone or combined with 1% PVA to maintain the viability of HCE-2 cells for further assays. 

Polyvinyl alcohol (PVA) is usually found in ophthalmic solutions as a lubricant to prevent irritation or to relieve dryness of the eyes because of its biocompatibility [46]. However, Vico et al. reported that PVA-treated dry eye patients did not show a significant improvement in any DES diagnostic parameters [47]. One of the effects of PVA is as a thickener to increase viscosity-enhancing eye drops retained on the ocular surface. As shown in the viscosity test (Table 1), the viscosity value increased when 1% PVA was added to the AT/P1 or AT/L5P1 groups. The physical properties, such as pH, osmolarity, and refractive index, in all tested groups were similar to those of normal human tears (Table 1). The addition of PVA effectively increased the ocular surface retention time of AT/L5P1 (Figure 4). However, PVA did not decrease the expression of inflammatory cytokine genes in HCECs (Figure 3). The results demonstrated the role of 1% PVA addition to a thickener/lubricant to retain AT on the eye with no effect on inflammatory inhibition. 

Tear volume reduction is a significant feature of DES that causes ocular dryness and ocular burning. The tear-secretory function can be disrupted by disease of the lacrimal functional unit, as well as from the ocular surface or glandular inflammation [48]. A BAC-induced DES model stimulates the overexpression of inflammatory cytokines, which may promote apoptosis of both epithelial and goblet cells, and further impairs tear film stability. The breakdown of the tear film aggravates ocular surface damage in turn [49]. To date, no study has reported the effect of lutein on conjunctival goblet cells. Meurer et al. found that the *Tagetes erecta* extract, which is rich in lutein, could inhibit the depletion of colon goblet cells [50]. It was speculated that lutein might also influence the conjunctival goblet cells. In our animal model of the BAC-induced DES group, tear secretion was significantly decreased compared to that in the control group (Figure 5). Fluorescein staining revealed severe corneal damage after BAC treatment in the DES group (Figure 6). Histological analysis of the cornea and conjunctiva by H&E and PAS staining also showed that the corneal epithelium became thinner, accompanied by inflammatory cell infiltration in the stroma (Figure 7), as well as loss of conjunctival goblet cells (Figure 8). In contrast, topical treatment with lutein eye drops administered for 10 days resulted in an effective recovery, indicated by the histological morphology. Although both AT/L5 and AT/L5P1 groups revealed similar conditions as the histological results, based on the results of ocular retention time (Figure 4), cornea intact (Figure 6), and inflammatory cytokine expression (Figure 9), the AT/L5P1 group showed better therapeutic effects in DES mice.

The pathophysiology of DES includes loss of tear film, tear hyperosmolarity, oxidative stress, and inflammation of the ocular surface that result in damage. Pro-inflammatory cytokines, such as IL-6, IL-8, and TNF-α, are high in patients with DES [51]. In addition to functioning as a blue-light filter, lutein is likely to reduce reactive oxygen species (ROS) levels by inducing the activity of SOD antioxidant enzymes [52,53]. It has been shown that lutein can reduce the expression of inflammatory factors by inhibiting ROS, thereby protecting retinal pigment epithelium cells from light-induced damage [54,55]. Li et al. suggested that 20 μM lutein treatment can inhibit the expression of inflammatory factors in retinal Müller cells [24]. Muz et al. developed a multicomponent oral formulation with curcumin, vitamin D3, and lutein/zeaxanthin and tested it in a BAC-induced rat DES model. After administration of the formulation by oral gavage at a total dose of approximately 1.4 mg lutein per rat for 4 weeks, it was effective in alleviating the symptoms of dry eye by reducing oxidative stress and inflammation and restoring mucin levels [56]. This study revealed the possibility of oral antioxidant/anti-inflammatory agents, including lutein, for treating DES. Here, we elucidated the anti-inflammatory effects of lutein by in vitro and in vivo examinations. Our data showed that 5 μM lutein mixed with 1% PVA could effectively suppress IL-1β, IL-6, and TNF-α gene expression in the LPS-induced HCEC cells (Figure 3) and inflammatory cytokine levels in the corneas of BAC-induced DES mice (Figure 9). This reveals that the anti-inflammatory effect of lutein-containing eye drops with 1% PVA would improve the therapeutic management of DES. We proved in this study that dosing twice daily with lutein-containing eye drops that have a very low lutein concentration (only 5 μM) has a good therapeutic effect for DES treatment.

## 5. Conclusions

In summary, this study demonstrates that topical application of lutein at 5 μM mixed with thickener PVA at 1% (L5P1) was safe for use in HCECs. The expression of inflammatory cytokines, such as IL-1β, IL-6, and TNF-α, was significantly downregulated when HCECs were treated with L5P1. The characterization of AT containing L5P1 was similar to that of human tears such as pH, osmolarity, viscosity, and refractive index. This effectively enhanced the drug-retention time on the ocular surface. Topical administration of AT containing L5P1 (eye drops) in BAC-induced DES mice rescued tear production, facilitated corneal wound healing, suppressed corneal and conjunctival goblet cell loss, and decreased inflammatory cytokine expression. The result was comparable with the therapeutic effect of a commercial CsA agent for DES treatment. Lutein-containing eye drops directly working on the eye, not delivered by the gastrointestinal route, can be used as a DES therapeutic agent by inhibiting inflammatory conditions on the ocular surface. The addition of 1% PVA enhanced ocular retention, facilitating the bioavailability of lutein for the effective treatment of DES. Further studies to fulfill pharmacies’ regulations need to be evaluated, and its application in DES clinics will be possible in the future.

## Figures and Tables

**Figure 1 pharmaceutics-13-01801-f001:**
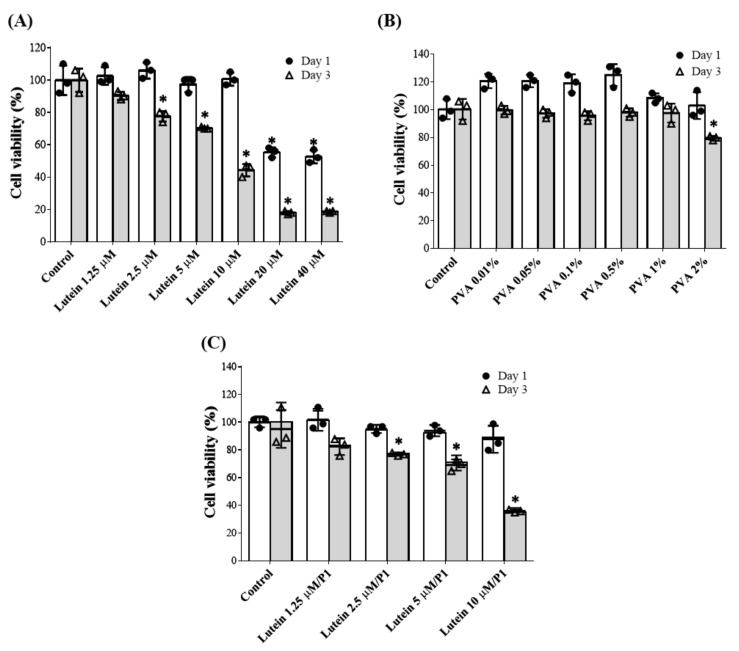
Cell viability of human corneal epithelial cells (HCE-2) upon coculture with varying concentrations of (**A**) lutein, (**B**) PVA, and (**C**) lutein/PVA combination for 1 and 3 days. Data are expressed as the mean ± standard deviation (SD) and analyzed by one-way ANOVA test; *n* = 6, (* *p* < 0.05 compared with the control group). P1: 1% PVA.

**Figure 2 pharmaceutics-13-01801-f002:**
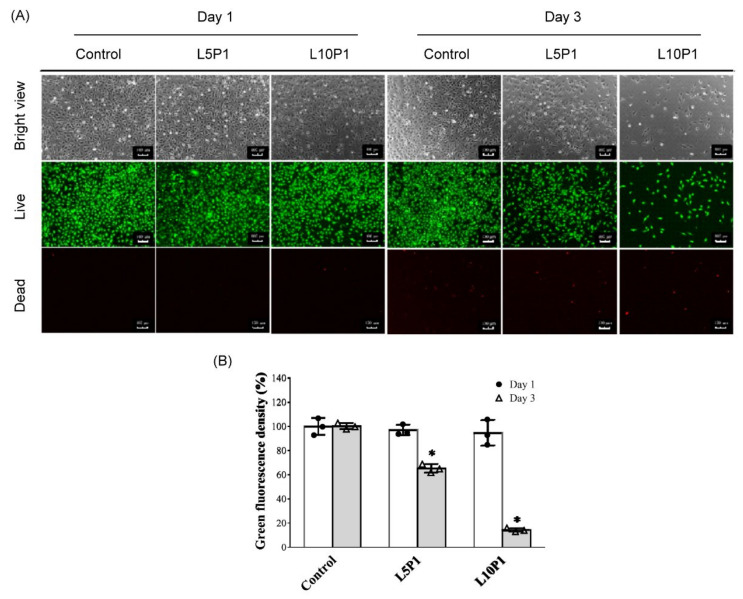
(**A**) Live/dead staining images of HCE-2 cells treated with L5P1 (5 μM lutein mixed 1% PVA) and L10P1 (10 μM lutein mixed 1% PVA) for 1 and 3 days. Green: live cells; red: dead cells (Scale bar: 100 μm). (**B**) Quantitation of green fluorescence from live/dead staining images; *n* = 3, (* *p* < 0.05 compared with the control group).

**Figure 3 pharmaceutics-13-01801-f003:**
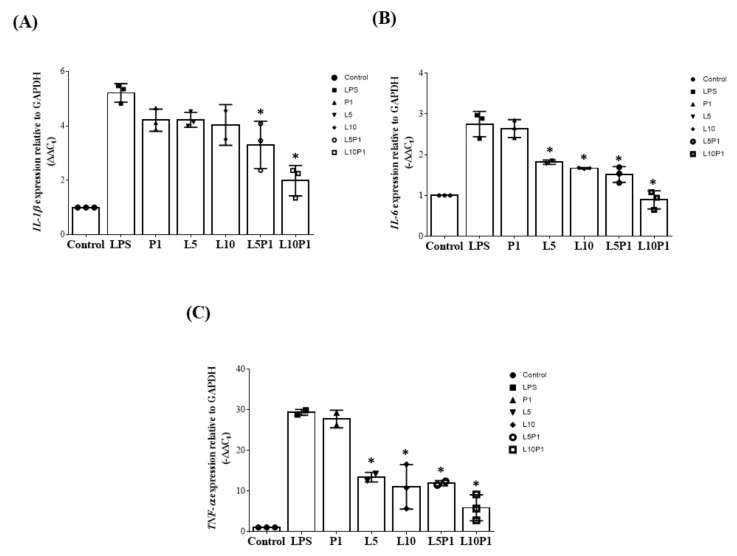
Expression of (**A**) IL-1β, (**B**) IL-6, and (**C**) TNFα in HCE-2 upon LPS-induced inflammation (6 h) and treatment with various lutein/PVA formulations for 2 h. The control group consisted of cells without LPS treatment. Results are displayed as the fold increase compared to the expression in normal HCE-2. All groups were compared with the LPS group for statistical analysis; *n* = 3, (* *p* < 0.05). LPS: lipopolysaccharide; L5: 5 μM lutein; L10: 10 μM lutein; P1: 1% PVA.

**Figure 4 pharmaceutics-13-01801-f004:**
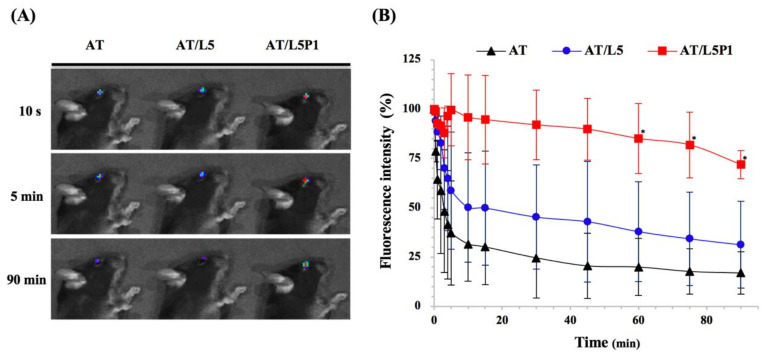
Retention test on the ocular surface. (**A**) Accumulation of fluorescent on mouse eyes after dosing with different formulations. (**B**) Quantitation of fluorescence intensity on the ocular surface traced by IVIS at different time intervals. Data are expressed as the mean ± standard deviation (SD); *n* = 3. (* *p* < 0.05 compared with the AT group).

**Figure 5 pharmaceutics-13-01801-f005:**
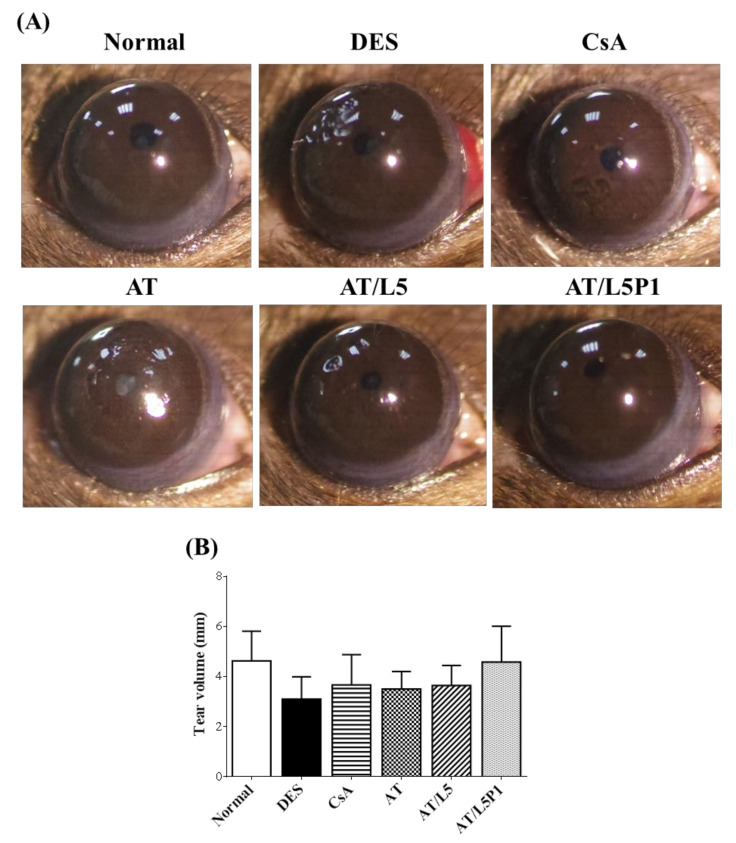
DES mice induced by BAC were treated with varying formulations of lutein and PVA, using eye drops as a topical delivery method for 10 days. (**A**) The appearance of mouse eyes. (**B**) Tear volume (Schirmer’s test) results. Data are expressed as the mean ± standard deviation (SD); *n* = 6.

**Figure 6 pharmaceutics-13-01801-f006:**
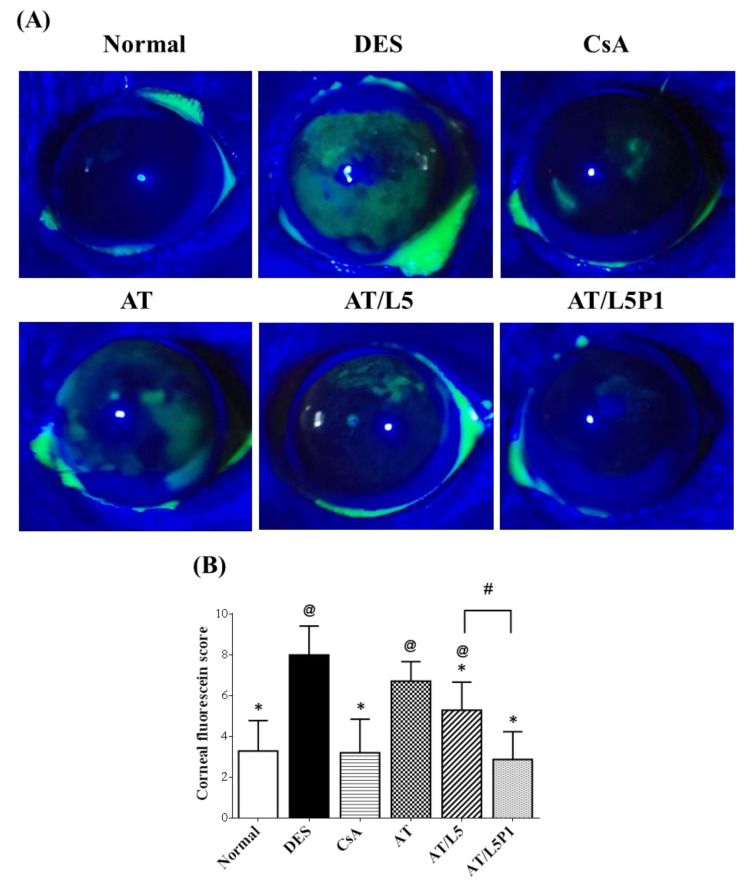
Corneal fluorescein staining of DES mice after 10 days treatment. (**A**) Slit-lamp photography of the mouse eyes in each group. (**B**) Cornea grading according to the National Eye Institute scale. Data were analyzed using the one-way ANOVA and are expressed as the mean ± standard deviation (SD); *n* = 6, (@ *p* < 0.05 compared with the control group, * *p* < 0.05 compared with the DES group, # *p* < 0.05 compared with the AT/L5P1).

**Figure 7 pharmaceutics-13-01801-f007:**
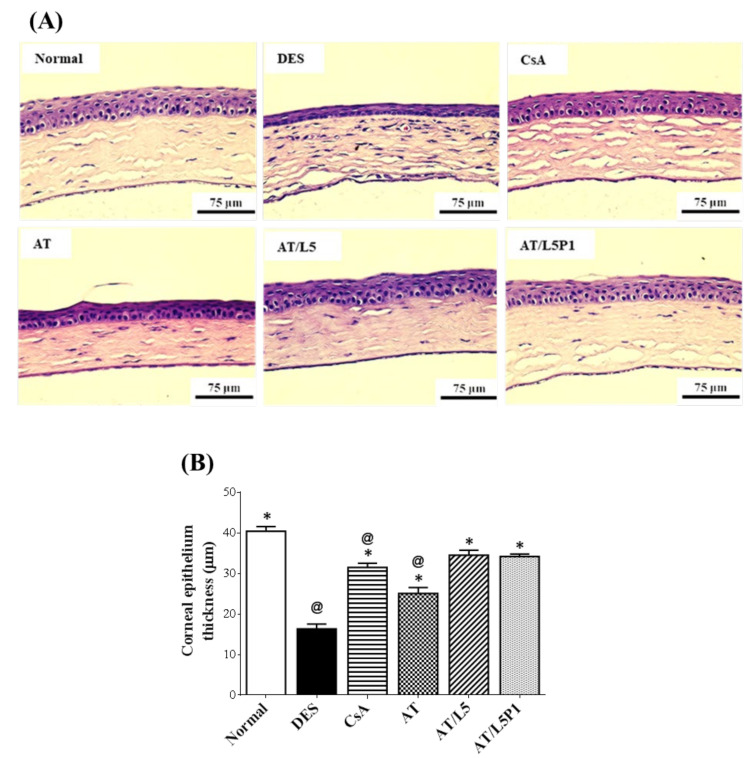
Effect of the combinatorial treatment with AT and lutein on the histopathology of BAC-induced dry eye syndrome in mice. (**A**) Hematoxylin and eosin (H&E) staining of corneal sections from the eyes of mice in various groups (Scale bar = 75 μm). (**B**) Quantification of corneal epithelial thickness. Data were analyzed using one-way ANOVA and are expressed as the mean ± standard deviation (SD); *n* = 5, (@ *p* < 0.05 compared with the control group, * *p* < 0.05, compared with the DES group).

**Figure 8 pharmaceutics-13-01801-f008:**
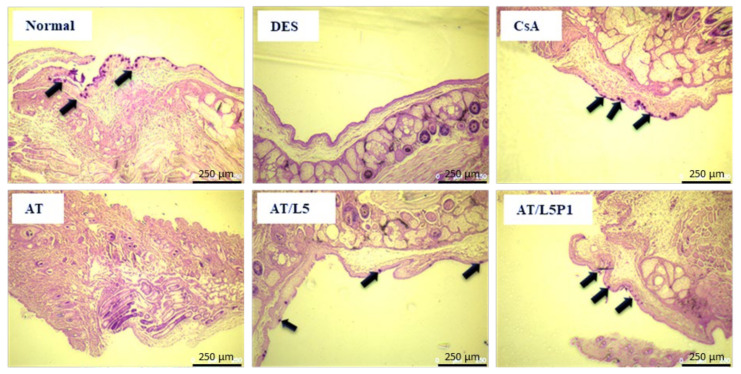
Microphotograph of conjunctival goblet cells followed by PAS staining of the eye sections from BAC-induced DES mice in variant groups (Scale bar = 250 μm). The arrows indicate the goblet cells.

**Figure 9 pharmaceutics-13-01801-f009:**
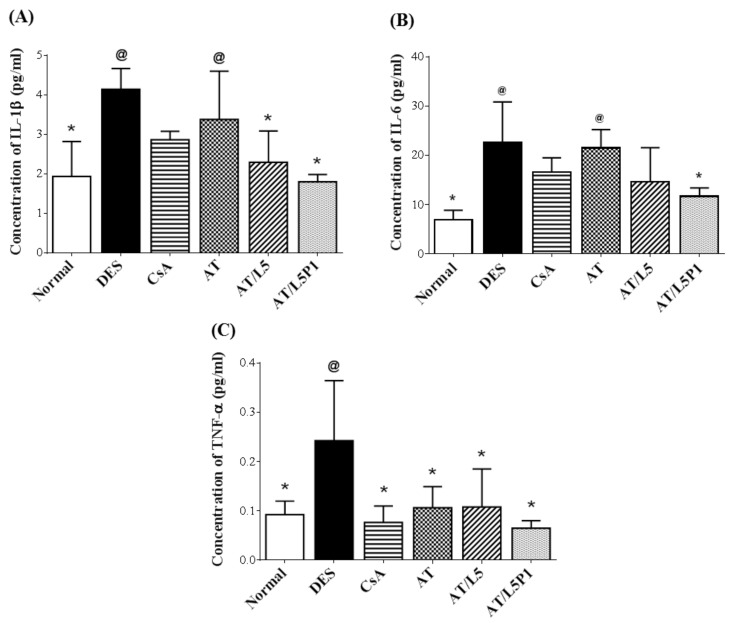
Quantification of inflammatory cytokines extracted from mouse corneas. Cytokine concentration of (**A**) IL-1β, (**B**) IL-6, and (**C**) TNF-α. Data were analyzed using the one-way ANOVA and are expressed as the mean ± standard deviation (SD); *n* = 6, (@ *p* < 0.05 compared with the control group, * *p* < 0.05 compared with the DES group).

**Table 1 pharmaceutics-13-01801-t001:** Characteristics of artificial tears (AT) with variant lutein and PVA combinations.

Group	pH Value	Osmotic Pressure (mOsm/kg)	Viscosity (mPa·s)	Refractive Index (RI)
Human tears	6.5~7.6 [31]	260~340 [32]	1~10 [33]	1.3369 ± 0.0011 [34]
AT	8.33 ± 0.22	253 ± 1	0.88 ± 0.03	1.3345 ± 0.0001
AT/L5	8.37 ± 0.01	261 ± 2	0.85 ± 0.11	1.3347 ± 0.0001
AT/P1	7.78 ± 0.01	263 ± 2	1.17 ± 0.05	1.3359 ± 0.0002
AT/L5P1	7.78 ± 0.01	271 ± 4	1.21 ± 0.02	1.3359 ± 0.0001

Data presented as mean ± standard deviation (*n* = 3). AT: artificial tears; L5: 5 μM lutein; P1: 1% PVA; L5P1: 5 μM lutein mixed with 1% PVA.

## Data Availability

Not applicable.

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
