# Peer review of "Development of Lutein-Containing Eye Drops for the Treatment of Dry Eye Syndrome"

_pharmaceutics, 2021, doi:10.3390/pharmaceutics13111801_

Round 1

Reviewer 1 Report

  1. Treatment for dry eye disease (DED) is aimed at improving symptoms by increasing or supplementing tear production, slowing tear evaporation, reducing tear resorption, or reducing ocular surface inflammation. The clinical importance of artificial tears as a first-line drug for dry eye treatment is not reflected in the manuscript.
  2. In the results of the study, the authors only emphasized the anti-inflammatory effects of luteolin and did not focus on the clinical outcomes, which led to inaccurate conclusions. The conclusions of this paper should be considered again cautiously.
  3. Please provide details of the sample size for each group and an explanation of the effect of sample size on the test results.
  4. Does the “n=3” at the bottom of Table 1 refer to the sample size? Please add it in the correct place and explain it.
  5. Abstract part, line 32, HCECs → human corneal epithelial cells (HCECs).
  6. Introduction part, line 88, LPS → lipopolysaccharide (LPS), delete the abbreviations below in line 95.
  7. 1 Materials and Reagents part, line 98, Keratinocyte Serum-Free Medium → Keratinocyte serum-free medium. And in line 117 “keratinocyte serum-free medium” could be replaced by the abbreviation KSFM. 
  8. 2.3 Live/Dead Staining part, line 136, “HCE-2 cells treated with various lutein concentrations and 1% PVA were stained with a live/dead staining kit (04511-1KT-F, Sigma-Aldrich) to observe live 137 cells.” “1%PVA” This data needs to be noted that it is derived from the results of the cell viability examination in the previous step.
  9. Line 149, delete (cDNA), which has already been noted in the previous part (line 103).
  10. Please explain: Why not use female mice in the experiment?
  11. How to decide the examination time point (1 and 3 days)?
  12. Please provide the Dot Plot of Figure 1, Figure 2B, and Figure 3.
  13. The quantity of mice in each group is small due to too many groups. It is recommended to further increase the number of samples in future experiments.

I recommend that accepting after modifying.

Reviewer 2 Report

Pharmaceutics 1400648.

In this manuscript authors explore the use of artificial tears containing poly-vinyl alcohol and Lutein as a possible treatment of Dry-eye syndrome. First, they show that Lutein concentrations below 10 mM have minimal effects on the viability of HCE-2 human corneal epithelial cells, as demonstrated by viability assays and the cytotoxic activity of lutein on this cell line. Moreover, the show that poly-vinyl alcohol does not have any cytotoxic effect on these cell. Furthermore, authors that HCE-2 cultures treated with Lutein had a decreased expression of the inflammatory cytokines IL-6, IL-1β, and TNF-α when the compound was applied to cell cultures together with poly-vinyl alcohol. Similar results were obtained when tear drops containing poly-vinyl alcohol and 5 mM Lutein were instilled into mouse eyes. Taking into account the results obtained with mice corneas which suggest a reduction on corneal damage induced by application of benzalkonium chloride as a model of dry eye syndrome, authors suggest that tear drops containing Lutein concentrations lower than 10 mM and 1% (v/v) poly-vinyl alcohol could be used for treatment of dry eye syndrome.

MAJOR ISSUES.

1). Results reported by authors are interesting and suggestive. However they should establish the effects of the treatment with Lutein and PVA for longer periods. They only report cell viability and cytotoxic effects of the mixture after treatment of cell cultures for three days. After such treatment they show that 5 mM Lutein and 1% PVA decrease about 20% cell viability, an effect which raises the question: Treatment for longer periods could be detrimental for corneal tissue? Although their assays in mice comprised 10 days suggesting that cytotoxic effect should be minimal, such a result could be explained by the continuous eye blinking and the production of tear film. It could be important to clarify this issue to remark the importance of their results.

2) Previous reports have described the application of Lutein on corneal surface as a possible agent to improve drug delivery (Liu et al., 2014, J Ophthalmol. 2014:304694), or to exert a photo protective effect against Ultraviolet-Induced Photokeratitis (Harada et al., 2017, Oxid Med Cell Longev. 2017:1956104). In the first work, researchers used nanocarriers made with lipids and loaded with 2-Hydroxypropyl-β-cyclodextrin and up to 7 mM Lutein. In the second authors orally administered Lutein or Astaxanthin. Although the second work did not applied Lutein directly onto ocular surface, both works might be considered antecedents for the present manuscript, and they were not cited. Please discuss whether some of these papers could be relevant for this manuscript.

Interestingly, Liu et al demonstrate that Lutein is accumulated, an mainly retained in cornea, although also reach part of the anterior segment in the eye, a fact that could explain the increased retention of the artificial tears on the ocular surface.
